# Photoluminescence Gas Sensing by Fluorescein-Dye Anions/1-Butanesulfonate/Layered Double Hydroxide Hybrid Materials under Humid Environment Conditions

**DOI:** 10.3390/nano11040914

**Published:** 2021-04-03

**Authors:** Ryo Sasai, Satoshi Yamamoto, Akane Naito

**Affiliations:** Department of Materials Chemistry, Graduate School of Natural Science and Engineering, Shimane University, 1060 Nishi-kawatsu-cho, Matsue, Shimane 690-8504, Japan; shanbenyu8@gmail.com (S.Y.); n_ichigomilk_a@yahoo.co.jp (A.N.)

**Keywords:** layered double hydroxide, photoluminescence gas sensing, humid condition

## Abstract

In this study, we investigated the photoluminous spectroscopic behavior of hybrid powder incorporating both anionic fluorescein dye (AFD) and 1-butanesulfonate (C4S) with layered double hydroxide (LDH) in the presence of NH_3_ or NO_2_ gas under various relative humidity conditions. In the presence of NH_3_ gas, drastic photoluminescence enhancement from the LDH/AFD/C4S hybrid was observed at relative humidity (RH) ≥ 40% when the NH_3_ reached a certain concentration. Meanwhile, the LDH/AFD/C4S hybrid was exposed to NO_2_ gas at various relative humidity conditions, and the following behavior was observed: At RH ≥ 60%, the photoluminescence (PL) intensity from the hybrid gradually decreased as NO_2_ concentration increased. Therefore, the LDH/AFD/C4S hybrid investigated in this study is inferred to be suitable for optical NH_3_/NO_2_ sensor devices, which can be used in humid air.

## 1. Introduction

With the development of Internet of Thing (IoT) in recent years, the demand for various sensors has been increasing. In particular, sensors that can selectively detect specific molecules in various media, such as in the atmosphere or water, have been attracting attention from academics and manufacturers, and research and development of materials for that purpose have been intensifying all over the world. As some of the most promising materials that can be used in these sensors, sensing materials that use the 2-dimensional interlayer spaces of ion-exchangeable layered inorganic compounds, as introduced in reviews, have been investigated by researchers [1,2,3,4,5,6,7,8,9,10,11,12,13,14]. For example, Yan et al. reported the detection of the adsorption of aromatic compounds by a quartz crystal microbalance (QCM) device that uses an organically modified clay film [9]. Specific molecules such as water and ammonium have also been detected through electrochemical signal changes, using clay-hosted hybrid materials with either manganese oxide [10] or polyurethane [11] as the electrode. Yamagishi et al. demonstrated that it is possible to create a clay interlayer space in which the chirality of molecules can be recognized [12], whereas Fujimura et al. reported that a cationic magnesium porphyrin incorporated in the clay interlayer space could show color-tone changes depending on the relative humidity [13]. On the other hand, Shi et al. reported that fluorescein intercalated in layered double hydroxide exhibited a rapid optical response depending on the pH of the aqueous solution [14]. Researchers have also investigated the photoluminescence (PL) properties of luminous dyes incorporated into ion-exchangeable layered inorganic compounds modified by amphiphilic molecules in the presence of various molecules [15,16,17,18,19,20,21]. For example, for rhodamine 6G dyes, the changes in PL with respect to the change in wetness, when these dyes were incorporated into laponite modified by alkyltrimethylammonium, were found to depend on the alkyl-chain length [15]. This phenomenon can be explained by the difference in the amount of hydrated water in the interlayer space modified by alkyltrimethylammonium cations, and the effect of hydrated water on the aggregation behavior of rhodamine 6G dyes. In the case of materials hybridizing both rhodamine 3B and alkyltrimethylammonium with lepidocrocite-type layered titanate, the PL color varies with water adsorption because of the difference in the strengths of electrostatic interaction between rhodamine 3B and the anionic site on the titanate nanosheet surface [16]. This hybrid material also exhibited remarkable PL quenching when exposed to humid basic gases such as NH_3_ and amine derivatives [16]. This behavior can be explained by the intramolecular cyclization reaction from cationic rhodamine 3B (luminous type) to lactone-type rhodamine 3B (non-luminous type), which could have been induced by increases in pH around the rhodamine 3B molecules incorporated in interlayer space. Meanwhile, the p*K*_a_ values of cationic free-base porphyrin were reported to be remarkably varied by intercalation into the interlayer of layered α-zirconium phosphate [17]. Researchers also investigated the PL properties of anionic fluorescein dye (AFD) incorporated into the interlayer space of layered double hydroxide (LDH) with 1-butanesulfonate (C4S) under various relative humidity (RH) conditions. In the dried state, the LDH/AFD/C4S hybrid did not exhibit a PL spectrum because the AFD molecules in the hybrid exist as lactone-type without luminescence ability [18,19,20,21]. On the other hand, when the relative humidity (RH) exceeded 30%, the AFD in the LDH/AFD/C4S hybrid was observed to exhibit photoluminescence. In particular, the PL intensity linearly increased as the RH increased. This phenomenon was caused by increases in the abundance ratio of dianion-type AFD with PL ability and lactone-type AFD without PL ability due to changes in the RH. This conversion reaction from lactone-type to dianion-type AFD induced by water adsorption was caused by pH changes from neutral to basic. Thus, when basic or acidic gases are adsorbed onto the LDH/AFD/C4S hybrid, the PL properties of the hybrid are expected to change according to the amount of gas adsorbed. In this study, we investigated the PL characteristics of an LDH/AFD/C4S hybrid when ammonia, which is a basic gas, or nitrogen dioxide, which is an acidic gas, was adsorbed under various RH conditions.

## 2. Materials and Methods

### 2.1. Materials

Layered double hydroxide (LDH: [Mg_3_Al(OH)_8_]CO_3_∙*n*H_2_O, anion exchange capacity (AEC): 3.25 mmol/g) was gifted by Kyowa Chemical Industry Co. Ltd. Anion fluorescein dye (AFD; Tokyo Chemical Industry Co. Ltd. Tokyo, Japan) was used as the fluorescent dye, without further purification. Sodium 1-butanesulfonate (C4S: Tokyo Chemical Industry Co. Ltd.) was used as the modifier of the LDH interlayer space, without further purification.

### 2.2. Photoluminescence Gas-Sensing Measurement under Various Relative Humidity Environments

The LDH/AFD/C4S hybrid powder was prepared according to a procedure described in previous papers [14,16]. The chemical composition of the prepared LDH/AFD/C4S hybrid powder, which had a greenish-yellow color, was [Mg_3_Al(OH)_8_](AFD)_0.00025_(C4S)_0.69_(CO_3_)_0.15_∙2.98H_2_O.

The PL spectra of samples of the LDH/AFD/C4S hybrid powder were measured under a variety of NH_3_ or NO_2_ (abbreviated as [NH_3_] or [NO_2_]) concentrations and relative humidities (RH), using a custom-made in-situ PL measurement system (cf. Figure 1) according to the following procedures: (1) Each LDH/AFD/C4S hybrid powder sample was placed into a sample holder made of silicone, which was then placed inside a sample quartz cell. (2) The LDH/AFD/C4S hybrid powder sample was treated under dry N_2_ gas flow. (3) The PL spectrum of the dried LDH/AFD/C4S hybrid powder sample excited by a UV-LED irradiation apparatus (λ = 365 nm, KEYENCE, Tokyo, Japan) was recorded using a multichannel fiber spectrophotometer (Ocean Photonics). (4) N_2_ gas with various RH values (~80%), which were measured using a digital hygrometer, was produced via passage of dry N_2_ gas through a pure water medium. (5) The LDH/AFD/C4S hybrid powder sample was kept under a humid N_2_ gas flow at a given RH. (6) After the water-adsorption equilibrium state was achieved, the PL spectrum of the LDH/AFD/C4S hybrid powder sample was measured. (7) NH_3_ or NO_2_ gases with a given RH were prepared via combination with dried NH_3_ or NO_2_ gases ([NH_3_]~6.0 ppm, [NO_2_] = 25 ppm), which were generated by a calibration gas-generation system with a permeation tube (PD-18, GASTEC CORPORATION) using dried N_2_ gas as the carrier gas and humid N_2_ gas. (7) The prepared mixed gas was injected into the quartz cell sample. (8) The PL spectrum of the LDH/AFD/C4S hybrid powder sample was measured.

## 3. Results and Discussion

The PL spectra of the LDH/AFD/C4S hybrid powder sample under humid NH_3_ gas flow conditions (RH = 60%, [NH3] = 5.33 ppm) at a given exposure time are shown in Figure 2. According to the figure, the PL was promptly enhanced when the LDH/AFD/C4S hybrid powder sample was exposed to NH_3_ gas. At the same time, no change in the shape of the PL spectra could be observed; thus, this enhancement can be explained by the increase in the amount of fluorophore species in the LDH/AFD/C4S hybrid powder sample. In the inset of Figure 2, the PL intensity at 525 nm is plotted against exposure time. The PL intensity at 525 nm was observed to rapidly increase when the LDH/AFD/C4S hybrid powder sample was exposed to humid NH_3_ gas.

In Figure 3a, the PL enhancement factor *E*_RH_ due to the relative humidity, as defined in Equation (1), was plotted against the RH values.
*E*_RH_ = (*I*_RH_ − *I*_RH=0_)/*I*_RH=0_,(1)
where *I*_RH=0_ and *I*_RH_ are the PL intensities under dry and humid conditions, respectively. In Figure 3b–e, the PL enhancement factor *E*_NH3_ due to NH_3_, as defined in Equation (2), was plotted against [NH_3_] at various RH values.
*E*_NH3_ = (*I*_NH3_ − *I*_NH3=0_)/*I*_NH3=0_,(2)
where *I*_NH3=0_ and *I*_NH3_ are the PL intensities without NH_3_ and at a given [NH_3_], respectively.

Below RH = 20%, no *E*_NH3_ value was observed, even when [NH_3_] was increased. For comparison, in a previous study [16], the following were clearly observed and inferred: (1) AFD dyes in the LDH/AFD/C4S hybrid powder exist in the lactone form (see Scheme 1a) and have no PL ability under dry conditions. (2) Although water adsorption by the LDH/AFD/C4S hybrid powder occurred up to RH = 20%, AFD dyes in the hybrid powder still existed in the lactone form (see Scheme 1a). (3) Most of the adsorbed water molecules were localized on the surface of the metal hydroxide layer at RH < 20%. (4) The adsorbed water molecules did not affect the chemical state of AFD at RH < 20%. (5) PL enhancement, which started to be observed at RH > 30%, was caused by the conversion reaction from lactone form to dianion form via monoanion form, because the pH around the AFD dyes increased as an effect of the adsorption of water around the dyes. Based on these experimental inferences, the reason for no PL enhancement being observed at RH < 20% was the lack of adsorbed water molecules around the AFD dyes. On the other hand, when RH was set to 40%, PL enhancement of the LDH/AFD/C4S hybrid powder began to be observed, specifically when the [NH_3_] value was at least 1.2 ppm. Furthermore, *E*_NH3_ was saturated to approximately 0.2 at [NH_3_] ≈ 2.0 ppm (see Figure 3b). When RH was increased up to 60%, the [NH_3_] at which PL enhancement began to be observed ([NH_3_]^threshold^) shifted to 1.7 ppm, and the saturated *E*_NH3_ (^S^*E*_NH3_) also increased up to 0.4, which was twice the ^S^*E*_NH3_ value at RH = 20% (see Figure 3c). When RH was further increased to 80%, [NH_3_]^threshold^ became 3.1 ppm, and ^S^*E*_NH3_ increased up to 1.2 (see Figure 3d). Based on these findings and in conjunction with the previous study [16], these PL enhancements were deduced to have been caused by an increased abundance of the dianion form as a result of increases in pH due to the dissolution of ammonia in the water surrounding the AFD in the LDH/AFD/C4S hybrid powder. According to these results, the tested LDH/AFD/C4S hybrid powder has the capability to detect ammonia in air under humid conditions (RH ≥ 40%).

As shown in Figure 4a, [NH_3_]^threshold^ increased linearly with increases in RH. Because the conversion reaction from the lactone and/or monoanion forms to the dianion form of AFD does not proceed unless [NH_3_] exceeds a certain value, the adsorbed amount of ammonia required to produce saturated ammonia water will increase as RH increases, because the amount of water molecules around the AFD molecules increases as RH increases, as has been reported. Thus, the lower limit for the detection of ammonia by the tested LDH/AFD/C4S hybrid powder decreased as RH increased and was approximately 3 ppm at higher humidities (RH ≥ 80%). In particular, ^s^*E*_NH3_ increased linearly as RH increased (Figure 4b). As shown in a previous report [2], the PL enhancement by the changes in RH was caused by the conversion of both the lactone and monoanion forms to the dianion form in the LDH/AFD/C4S hybrid powder. As a result, the amount in dianion form increased, whereas the amounts in both the lactone and monoanion forms decreased, as the RH increased. However, part of the AFD still existed in monoanion form even when the RH was more than 80%, at which point *E*_NH3_ was at its saturated value. When the LDH/AFD/C4S hybrid powder is in the aforementioned state at a given RH, the reason for the increase in ^s^*E*_NH3_ as the RH increases is that the amount in dianion form will be higher at RH = 80% than at RH = 40%. This is because the abundance of the monoanion form, which is changeable to dianion form, will increase as RH increases. According to these experimental findings, this LDH/AFD/C4S hybrid powder reveals the presence of a specific amount of ammonia in moist space by enhancing PL; that is, this powder can be used as an ammonia detection material at high humidities.

In Figure 5b–e, the PL enhancement factor *E*_NO2_, which is defined in Equation (3), was plotted against [NO_2_] at various RH values.
*E*_NO2_ = (*I*_NO2_ − *I*_NO2=0_)/*I*_NO3=0_,(3)
where *I*_NO2=0_ and *I*_NO2_ are the PL intensities without NO_2_ and at a given [NO_2_], respectively. At RH = 0.1% (dry state), no *E*_NO2_ could be observed even when the LDH/AFD/C4S hybrid powder sample was placed in the NO_2_ gas flow. As shown in a previous report [16], the dependence of *E*_NO2_ on [NO_2_] at RH = 0.1% could be the reason that all the AFD dyes exist in the lactone form in the LDH/AFD/C4S hybrid powder. When the hybrid powder is exposed to NO_2_ gas at RH = 40%, PL quenching is expected to occur via the conversion of the dianion form to the lactone form because of the decrease in pH due to the production of nitric acid. However, no PL quenching was observed. The reason for this unexpected experimental finding is that the amount of nitric acid produced around the lactone form of the AFD dye was insufficient to lower the pH. Meanwhile, a gradual decrease in *E*_NO2_ was observed with increases in [NO_2_] at RH ≥ 60%. As shown in a previous report [16], the hydrated water around the AFD dye exists as a basic aqueous solution, and most AFD dyes will be in dianionic form. When NO_2_ molecules are dissolved in the basic hydrated water of the LDH/AFD/C4S hybrid powder, a neutralization reaction will occur around the AFD dye in the hybrid powder, and as a result, the pH will gradually decrease.

## 4. Conclusions

The photoluminous spectroscopic behavior LDH/AFD/C4S hybrid powder was investigated in the presence of NH_3_ or NO_2_ gas under various relative humidity conditions. In the presence of NH_3_ gas, PL intensity from the LDH/AFD/C4S hybrid was drastically enhanced at RH ≥ 40%, when the NH_3_ reached a certain concentration. These results indicated that the LDH/AFD/C4S hybrid powder has potential for NH_3_ detection in humid air (RH ≥ 40%). In particular, the present LDH/AFD/C4S hybrid has the potential to quantitatively detect ammonia in the concentration range of 1 to 4 ppm, which is equivalent to the human sense of ammonia smell, and is expected to be applied to the fields of welfare and long-term care. Meanwhile, when the LDH/AFD/C4S hybrid was exposed to NO_2_ gas at various relative humidity conditions, the following behavior was observed: At RH ≥ 60%, the PL intensity from the hybrid gradually decreased as NO_2_ concentration increased. In the case of NO_2_ gas exposure, the LDH/AFD/C4S hybrid was determined to be able to quantitatively detect NO_2_ in the high concentration range of 5 to 25 ppm in humid air (RH ≥ 60%). From these results, the present LDH/AFD/C4S is expected to be applied to the detection of NO_2_ in exhaust gas, which requires the detection of high-concentration NO_2_ in humid air (RH ≥ 60%). Therefore, the LDH/AFD/C4S hybrid investigated in this study is inferred to be suitable for optical NH_3_/NO_2_ sensor devices, which can be used in humid air. Moreover, this LDH/AFD/C4S hybrid can be used on various toxic molecules in air or water, which is very interesting for us. In future research endeavors, we can investigate the sensing ability of the LDH/AFD/C4S hybrid on tiny amounts of various toxic molecules in air or water.

## Data Availability

Data available in a publicly accessible repository that does not issue DOIs.

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
