# Peer review of "Photoluminescence Gas Sensing by Fluorescein-Dye Anions/1-Butanesulfonate/Layered Double Hydroxide Hybrid Materials under Humid Environment Conditions"

_nanomaterials, 2021, doi:10.3390/nano11040914_

Round 1

Reviewer 1 Report

The authors present interesting new results describing behavior of powder sensor LDH/AFD/C4S sensitivity to NH3/NO2 gas. The authors analyzed the sensitivity of mentioned structure under wide range of humidity values. The obtained results are very promising, especially at range RH ≥ 60%. However, strong decreasing of photoluminescence enhancement factor under RH =40% makes questionable practical usage of the sensor under ambient environment.  

To my opinion the text is clear and well organized. The results presented in clear form and explained well.  I have no significant objection

However, there are few points which need to be clarified:

  1. How long was the sample treated with dry N2 gas? Was the sample dried after each measurement?
  2. Figure 2. Three are two curves at the main plot. It is not clear what each of them corresponds to.

Author Response

Reviewer 1

  • How long was the sample treated with dry N2 gas? Was the sample dried after each measurement?

【Answer】

The sample was dried until the PL spectrum did not change. In addition, the sample was dried for each measurement under each humidity.

  • Figure 2. There are two curves at the main plot. It is not clear what each of them correspond to.

【Answer】

Thank you for pointing out. According to reviewer’s comment, I revised the Figure 2 and figure captain of Figure 2.

Reviewer 2 Report

In "Photoluminescence Gas Sensing by Fluorescein-Dye Anions/1- Butanesulfonate/Layered Double Hydroxide Hybrid Materials under Humid Environment Conditions" authors investigated the sensing properties of hybrid powder in high humidity atmosphere with respect to NH3 and NO2. Humidity is essential to activate the sensing mechanism based on enhancement or quenching of fluorescence, in case of NH3 and NOrespectively. The work is surely interesting even if in the reviewer opinion some points should be commented or improved (vide infra) in order to better understand the potentialities of this material as sensing material.

1) Why do the authors perform measurements in nitrogen atmosphere? Can the results obtained be reproduced also in synthetic air atmosphere?

2) Concerning the sensing performances, it seems that the "sensor" reaches saturation in a very short interval of concentration. Thus, to the reviewer it seems to be more an ON/OFF indicator which reveals the presence of analyte above a certain threshold ( ppm or tens of ppm for NH3 and NOrespectively.) It would be interesting having a comment on this aspect along with a list of potential applications for the detection of analytes in the concentration range founded as saturation limits.

3) The fluorescence emission is strongly affected by RH, which saturated vapor pressure depends on the temperature ,in turn. Do the author investigated the effect of temperature on the sensor responses? In case of a fixed RH value (e.g. 80%), do the emission  intensity changes with the temperature?

4)  Figure 2, please make evident in the figure the signal with and without [ NH3] by changing color or line style to the plot.

Author Response

Reviewer 2

  • Why do the authors perform measurement in nitrogen atmosphere? Can the results obtained be reproduced also in synthetic air atmosphere?

【Answer】

As you said, I understand that measurement in air is important for practical use as sensor, but I could not measure it because the gas concentration would be inaccurate if the carrier gas of the gas generator was change to air. In the future, I’d like to try measurement under moist air.

  • Concerning the sensing performances, it seems that the “sensor” reaches saturation in a very short interval of concentration. Thus, to the reviewer it seems to be more an ON/OFF indicator which reveals the presence of analyte above a certain threshold (ppm or tens of ppm for NH3 and NO2 respectively). It would be interesting having a comment on this aspect along with a list of potential applications for the detection of analytes in the concentration range founded as saturation limits.

【Answer】

Thank you for pointing out. As you said, we also think that the present LDH/AFD/C4S hybrid is an ON/OFF indicator that reveals it existence, especially for ammonia under humid conditions. On the other hand, regarding nitrogen dioxide, I think LDH/AFD/C4S hybrid can be used as a quantitative indicator in a certain concentration range, too.

  • The fluorescence emission is strongly affected by RH, which saturated vapor pressure depends on the temperature, in turn. Do the author investigated the effect of temperature on the sensor responses? In case of a fixed RH value (e.g. 80%), do the emission intensity changes with the temperature?

【Answer】

As you know, the PL is strongly dependent on temperature. We have not investigated the effects of temperature, so we cannot accurately mention the temperature effects. On the other hand, it is known that the PL of organic dyes decreases with increasing temperature because of molecular thermal vibration, so it is considered that the PL of the present LDH/AFD/C4S hybrid also decreases with an increase in temperature regardless of humidity.

  • Figure 2, please make evident in the figure the signal with and without [NH3] by changing color or line style to the plot.

【Answer】

Thank you for pointing out. According to reviewer’s comment, I revised the Figure 2 and figure captain of Figure 2.

Round 2

Reviewer 2 Report

The reviewer acknowledges that authors cannot perform additional experiment about air environment or temperature effect, although it should be appropriate if the authors may include in the manuscript some potential applications for the materials considering the threshold and linear ranges founded in the work. For example, NH3 linear range is [1-4] ppm ( usually exposure limit to ammonia is 25 ppm, far high than saturation region of the sensor); and in case of NO2 the linear range is [5-25] ppm whereas actual exposure limits are fixed at 1 ppm. It would greatly improve the significance of manuscript if authors may include a comment about the applications of these material as chemical sensor on the basis of the results obtained.

Author Response

Reviewer 2

  • The reviewer acknowledges that authors cannot perform additional experiment about air environment or temperature effect, although it should be appropriate if the authors may include in the manuscript some potential applications for the materials considering the threshold and linear ranges founded in the work. For example, NH3 linear range is [1-4] ppm ( usually exposure limit to ammonia is 25 ppm, far high than saturation region of the sensor); and in case of NO2 the linear range is [5-25] ppm whereas actual exposure limits are fixed at 1 ppm. It would greatly improve the significance of manuscript if authors may include a comment about the applications of these material as chemical sensor on the basis of the results obtained.

【Answer】

Thank you for your suggestions for further enhancing the effectiveness of my manuscript. Ammonia can be sensed in the same way as the human sense of ammonia smell, so it may be applied to the fields of welfare and long-term care. On the other hand, nitrogen dioxide can be applied to NO2 detection in exhaust gas. The following text has been added to the Conclusion section to indicate the fields in which the present LDH/AFD/C4S hybrid shown in my manuscript can be applied.

Line 206, “… in humid air (RH ≥ 40%). In particular, the present LDH/AFD/C4S hybrid has the potential to quantitatively detect ammonia in the concentration range of 1 to 4 ppm, which is equivalent to the human sense of ammonia smell, and is expected to be applied to the fields of welfare and long-term care. Meanwhile, when …”

Line 208, “… decreased as NO2 concentration increased. In the case of NO2 gas exposure, the LDH/AFD/C4S hybrid was determined to be able to quantitatively detect NO2 in the high concentration range of 5 to 25 ppm in humid air (RH ≥ 60%). From these results, the present LDH/AFD/C4S is expected to be applied to the detection of NO2 in exhaust gas, which requires the detection of high-concentration NO2 in humid air (RH ≥ 60%). Therefore, the …”